# Magnetic Monopole Molecules

Kazuyuki Furuuchi[†] and Moreshwar Pathak

*Manipal Centre for Natural Sciences,*
*Manipal Academy of Higher Education,*
*Manipal 576 104, Karnataka, India*

## Abstract

We construct a variety of bound states of Dirac magnetic monopoles in product $U(1)$ gauge theories that make up a Dirac magnetic monopole with unit magnetic charge under the unbroken $U(1)$ gauge group. The size of the bound states is determined by the balance between the repulsive magnetic Coulomb force of the unbroken $U(1)$ gauge group and the attractive force from the tension of the magnetic flux tubes of the broken $U(1)$ gauge groups. These bound states are extensions of the configuration first studied in [1]. We dub this type of configurations "Magnetic Monopole Molecules" (MMMs). Besides some illustrative examples of MMMs made of a small number of constituent Dirac magnetic monopoles, a method to combine smaller MMMs to construct larger MMMs is presented. Implications for the weak gravity conjecture are also discussed.

---

[†]Corresponding author.

# 1 Introduction

Magnetic monopoles [2, 3]; [4, 5] occupy a special place in the study of gauge theories. They are the key ingredients of the electric-magnetic duality [2, 3, 6, 7], and they play important roles in the understanding of non-perturbative aspects of gauge theories [8, 9]. Experimentally, the non-detection of magnetic monopoles put important constraints on models of the early universe [10, 11, 12, 13, 14].

Magnetic monopoles also played key roles in the recent studies on the swampland programme [15], in particular, in the magnetic version of the weak gravity conjecture (WGC) [16]. Swampland conjectures are proposed criteria that select out effective field theories (EFTs) inconsistent with quantum gravity. While studying whether the constraints from the WGC descend to low energy, Saraswat found a composite Dirac magnetic monopole that is made of constituent Dirac magnetic monopoles, each from one of the $U(1)$ gauge groups in a high-energy theory [1]. The configuration of the constituent Dirac magnetic monopoles is stabilized by the balance between the repulsive magnetic Coulomb force of the unbroken $U(1)$ gauge group and the attractive force from the tension of the Nielsen-Olesen type magnetic flux tubes [17] connecting the constituent Dirac magnetic monopoles [18]. Thus, for this kind of configurations, the existence of the unbroken $U(1)$ gauge group is crucial: the configuration in total has the unit magnetic charge of the unbroken $U(1)$ gauge group, where the unit magnetic charge is determined by the smallest unit of the electric charge in the theory, via the Dirac charge quantization condition [2]. The important roles of this type of composite magnetic monopoles in the WGC were further investigated in [19], while their roles in the lattice version of the WGC [20, 21] were investigated in [22]. The role of the composite magnetic monopoles in the swampland distance conjecture (DC) [23] through its possible connection to the WGC [24] was investigated in [25, 26]. Similar composite magnetic monopoles were studied [27] in dimensionally deconstructed gauge theories [28, 29].

The way multiple constituent Dirac magnetic monopoles are connected by the magnetic flux tubes in this type of configurations reminds us of molecules made of atoms. In fact, the low-energy excitation spectra of the composite magnetic monopoles were analyzed in [25, 26] in a way similar to what we do for studying excitation energy spectra of molecules. Moreover, as we show in this article, we can construct a larger composite magnetic monopole by assembling smaller composite magnetic monopoles, qualitatively similar to the way smaller molecules are combined to form a larger molecule. Therefore, we call such bound states of constituent Dirac magnetic monopoles to make up a unit charge Dirac magnetic monopole "Magnetic Monopole Molecules (MMMs)" in this article.

Configurations of magnetic monopoles connected by magnetic flux tubes were also studied in [10, 30]. The relevant gauge groups are all Higgsed in these configurations. In contrast, as mentioned above, the configurations studied in this article have a unit

magnetic charge of the unbroken $U(1)$ gauge group, and this is crucial for their stability. Another difference is that the configurations in [10, 30] were constructed in non-Abelian gauge theories, while our configurations are based on the Dirac magnetic monopoles in EFTs with $U(1)$ gauge groups and with a UV cut-off, although it would be interesting to embed (UV complete) the constituent Dirac magnetic monopoles in non-Abelian gauge theories. The solutions studied in [30] are BPS solutions [31, 32, 33] in supersymmetric gauge theories. While supersymmetry provides us with better theoretical control, it limits the stage to supersymmetric gauge theories. For the configurations studied here, we may not find exact solutions, but the existence and the stability of the bound states are guaranteed by the simple force balance argument and are robust. Therefore, these configurations are expected to have a broader appearance in non-supersymetric gauge theories and the supersymmetry-broken phase of supersymmetric gauge theories.

In this article, we first construct MMMs in the $N = 2$ case, where $N$ is the number of the $U(1)$ gauge groups. In this case, we make a list that is essentially complete, apart from minor variations.

In the $N = 3$ case we study next, we still cover a large class of MMMs, but the list is not complete. That we don't make a complete list may be quite expected, considering the rich variety of possible configurations, qualitatively in analogy to the real molecules made of atoms.

Rather than trying to make a complete list, we then take an approach to construct larger MMMs by assembling smaller MMMs, again, in good analogy to the real molecules made of atoms.

Finally, we study the gravitational stability of the MMMs from their internal structure in the context of the WGC. The result has interesting implications for the issue of how the constraints from a swampland conjecture on a high-energy EFT may descend to lower energy scales.

## 2   Magnetic Monopole (MMMs)

The model we consider has a product of $U(1)$ gauge groups $\prod_{i=1}^{N} U(1)_i$. As mentioned in the introduction section, the MMM configurations we consider have a unit magnetic charge of the unbroken $U(1)$ gauge group. The constituent Dirac magnetic monopoles are connected [18] by the magnetic flux tubes [17] of the broken gauge groups. The attractive forces from the tension of the magnetic flux tubes balance the repulsive magnetic Coulomb forces between the constituent Dirac magnetic monopoles. Therefore, both the broken part and the unbroken part of the gauge group are necessary for the stability of this type of configurations.

## 2.1  $N = 2$

We start with a slight extension of the model by Saraswat [1] with the gauge group $U(1)_1 \times U(1)_2$. For simplicity, we assume that both $U(1)$ gauge groups have the same gauge couplings, $g_1 = g_2 = g$. The charges of the particles and the Dirac magnetic monopoles of the model are listed in Table 1. The charged matter fields $\psi_1$ and $\psi_2$ are included just to define the smallest charge unit in each $U(1)$ gauge group. Their dynamical properties are not relevant in our discussions below. The Higgs field has charge $(-Y, X)$ under the $U(1)_1 \times U(1)_2$ gauge group, where we assume that the charges are quantized, i.e. $X$ and $Y$ are non-zero integers. $X = 1$ (or $Y = 1$) case essentially corresponds to the case originally studied in [1]. Without a loss of generality, we assume $0 < X \leq Y$. We further assume that $X$ and $Y$ are mutually prime: When $X$ and $Y$ have a common divisor, it is absorbed in the normalization of the gauge couplings of the unbroken and the broken $U(1)$ gauge groups, and the rest of the analysis is quite similar to the case when $X$ and $Y$ are mutually prime. Apart from such minor variations,[1] the list here covers all possible MMMs for the case $N = 2$.

It is convenient to associate ortho-normal basis vectors $\vec{e}_i$ $(i = 1, 2)$ to the respective $U(1)$ gauge groups:

$$\vec{e}_i \cdot \vec{e}_j = \delta_{ij} \,. \tag{2.1}$$

We can express the unit vectors in the unbroken and the broken $U(1)$ gauge groups as

$$\vec{e}_{\tilde{1}} = \cos\phi \, \vec{e}_1 + \sin\phi \, \vec{e}_2 \,, \tag{2.2}$$

$$\vec{e}_{\tilde{2}} = -\sin\phi \, \vec{e}_1 + \cos\phi \, \vec{e}_2 \,, \tag{2.3}$$

where

$$\cos\phi = \frac{X}{\sqrt{X^2 + Y^2}} \,. \tag{2.4}$$

Here, $\vec{e}_{\tilde{1}}$ is the basis vector associated to the unbroken $U(1)_{\tilde{1}}$ gauge group, and $\vec{e}_{\tilde{2}}$ is the basis vector associated to the broken $U(1)_{\tilde{2}}$ gauge group. The inverse transformation is given by

$$\vec{e}_1 = \cos\phi \, \vec{e}_{\tilde{1}} - \sin\phi \, \vec{e}_{\tilde{2}} \,, \tag{2.5}$$

$$\vec{e}_2 = \sin\phi \, \vec{e}_{\tilde{2}} + \cos\phi \, \vec{e}_{\tilde{2}} \,. \tag{2.6}$$

We normalize the gauge couplings so that the smallest unit of the charge in the theory is one. With this convention, when $Y$ and $Z$ are mutually prime which we assumed to be the case, the gauge couplings of the unbroken $U(1)_{\tilde{1}}$ gauge group and the broken $U(1)_{\tilde{2}}$ gauge group are given as

$$g_{\tilde{1}} = g_{\tilde{2}} = \tilde{g} := \frac{g}{\sqrt{X^2 + Y^2}} \,. \tag{2.7}$$

---

[1]For example, we may also have multiple Higgs fields with the same charge.

We impose the Dirac quantization condition [2] to the magnetic charges of the Dirac magnetic monopoles. Then, the smallest unit of the magnetic charges in the original $U(1)$ gauge groups $U(1)_1$ and $U(1)_2$ are

$$g_{m1} = g_{m2} = g_m := \frac{2\pi}{g} . \tag{2.8}$$

Following the smallest units of the electric charges in $U(1)_{\tilde{1}}$ and $U(1)_{\tilde{2}}$ gauge groups (2.7), the smallest units of the magnetic charges in $U(1)_{\tilde{1}}$ and $U(1)_{\tilde{2}}$ gauge groups are given as

$$g_{m\tilde{1}} = g_{m\tilde{2}} = \tilde{g}_m := \frac{2\pi}{\tilde{g}} = \frac{2\pi\sqrt{X^2 + Y^2}}{g} . \tag{2.9}$$

| particle | charge in $U(1)_1 \times U(1)_2$ | charge in $U(1)_{\tilde{1}} \times U(1)_{\tilde{2}}$ |
|---|---|---|
| $\psi_1$ | (e) $(1,0)$ | (e) $(X,-Y)$ |
| $\psi_2$ | (e) $(0,1)$ | (e) $(Y,X)$ |
| Higgs $H$ | (e) $(-Y,X)$ | (e) $(0, X^2 + Y^2)$ |
| $U(1)_1$ monopole | (m) $(1,0)$ | (m) $\left(\frac{X}{X^2+Y^2}, -\frac{Y}{X^2+Y^2}\right)$ |
| $U(1)_2$ monopole | (m) $(0,1)$ | (m) $\left(\frac{Y}{X^2+Y^2}, \frac{X}{X^2+Y^2}\right)$ |

Table 1: The case $N = 2$: (e) and (m) indicate the electric- and magnetic charge, respectively. $X$ and $Y$ are mutually prime integers. The charges are normalized so that the smallest charge units in the respective $U(1)$ gauge groups are one.

From Table 1. we observe that the Dirac magnetic monopole of the unbroken $U(1)_{\tilde{1}}$ gauge group with unit magnetic charge is made of $X$ constituent Dirac magnetic monopoles of $U(1)_1$ gauge group with a unit magnetic charge and $Y$ constituent Dirac magnetic monopoles of $U(1)_2$ with a unit magnetic charge. This set of constituent Dirac magnetic monopoles makes up what we call MMM.

The constituent Dirac magnetic monopoles are connected [18] by Nielsen-Olesen type magnetic flux tubes [17]. We observe the collection of $X$ $U(1)_1$ constituent Dirac magnetic monopoles and $Y$ $U(1)_2$ constituent Dirac magnetic monopoles has a zero net magnetic charge of the broken $U(1)_{\tilde{2}}$ gauge group, which means that a magnetic flux tube coming out of a constituent Dirac magnetic monopole will be absorbed in (an)other constituent Dirac magnetic monopole(s) of opposite magnetic charge. Note that possible ways to connect constituent Dirac magnetic monopoles by the magnetic flux tubes are not unique, generically, when $X$ or $Y$ are sufficiently large. The lowest energy configuration may be unique, but we will not attempt to find the lowest energy configuration in this work: We are content with the order of magnitude estimate of the ground state energy.

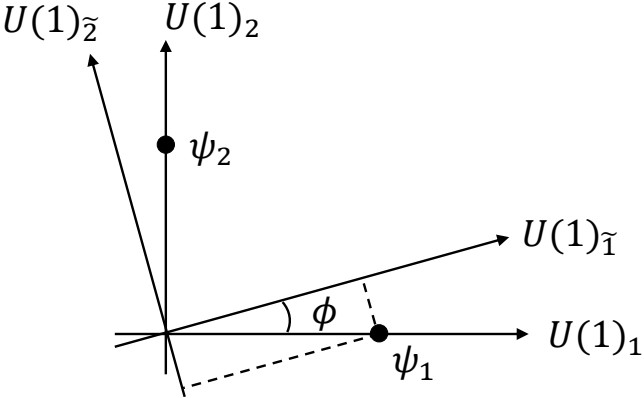

Figure 1: The relation between $U(1)_1 \times U(1)_2$ gauge group basis and $U(1)_{\tilde{1}} \times U(1)_{\tilde{2}}$ gauge group basis. The Higgs field has charges $(-Y, X)$ under the gauge group $U(1)_1 \times U(1)_2$, which is in the direction of the broken $U(1)_{\tilde{2}}$ gauge group. The angle $\phi$ is given by $\cos\phi = X/\sqrt{X^2 + Y^2}$. The charged fields $\psi_i$ ($i = 1, 2$) has the unit charge in respective gauge groups $U(1)_i$.

From Table 1, we observe that the Higgs field has $X^2 + Y^2$ units of electric charge in the broken $U(1)_{\tilde{2}}$ gauge group. However, the magnetic charges of the constituent Dirac magnetic monopoles in the broken $U(1)_{\tilde{2}}$ gauge group are fractional, with $X^2 + Y^2$ in the denominator. What is required for constructing magnetic flux tube solutions is that the magnetic flux in the unit with respect to the electric charge of the Higgs field is an integer [17, 27],[2] which is satisfied as we see in Table 1.

### 2.1.1 $N = 2$, $X = 1$, $Y = 1$: "Diatomic" MMM

It is instructive to first look at the simplest MMM of $N = 2$, $X = 1$, $Y = 1$, explicitly. A closely related configuration has been analyzed in some detail in [27]. By putting $X = 1$ and $Y = 1$ in Table 1, we obtain Table 2. For simplicity, We assume that the gauge couplings of the $U(1)_1$ gauge group and the $U(1)_2$ gauge group are equal, $g_1 = g_2 = g$. Then, following the Dirac quantization condition, the smallest unit of the magnetic charges in the original $U(1)$ gauge groups $U(1)_1$ and $U(1)_2$ are given as

$$g_{m1} = g_{m2} = g_m := \frac{2\pi}{g}. \tag{2.10}$$

---

[2]The vorticity of the magnetic flux tube is most conveniently measured in the unit of the electric charge $g_H$ of the Higgs field involved, or more precisely, the corresponding magnetic charge, $2\pi/g_H$. When the magnetic vortices repel each other, corresponding to the type II superconductor, the magnetic fluxes are distributed among unit vorticity magnetic flux tubes.

The gauge couplings in the unbroken $U(1)_{\tilde{1}}$ gauge group and the broken $U(1)_{\tilde{2}}$ gauge groups are given as

$$g_{\tilde{1}} = g_{\tilde{2}} = \tilde{g} := \frac{g}{\sqrt{2}} \,. \tag{2.11}$$

Following the smallest units of the electric charges in $U(1)_{\tilde{1}}$ and $U(1)_{\tilde{2}}$ gauge groups (2.11), the smallest units of the magnetic charges in the gauge couplings of the gauge groups $U(1)_{\tilde{1}}$ and $U(1)_{\tilde{2}}$ are given as

$$g_{m\tilde{1}} = g_{m\tilde{2}} = \tilde{g}_m := \frac{2\pi}{\tilde{g}} = \frac{2\sqrt{2}\pi}{g} \,. \tag{2.12}$$

It follows that the unit charge Dirac magnetic monopoles in the $U(1)_1$ and $U(1)_2$ gauge groups have fractional magnetic charges in the unbroken $U(1)_{\tilde{1}}$ gauge group, see Table 2. The unit charge Dirac magnetic monopole of the unbroken $U(1)_{\tilde{1}}$ gauge group can be constructed by combining one Dirac magnetic monopole of $U(1)_1$ gauge group and one Dirac magnetic monopole of $U(1)_2$ gauge group [1]. They repel each other by the magnetic Coulomb force of the unbroken $U(1)_{\tilde{1}}$ gauge group. However, the tension of the magnetic flux tube of the broken $U(1)_{\tilde{2}}$ gauge group balances the repulsive magnetic Coulomb force. A schematic figure of the "diatomic" MMM is given in Fig. 2. The balance determins the size of the MMM [1, 19]:

$$L_{\mathrm{MMM}} \simeq \frac{1}{gv} \,, \tag{2.13}$$

where $v$ is the vacuum expectation value (VEV) of the Higgs field. At the distance sclaes shorter than $L_{\mathrm{MMM}}$, the EFT of $U(1)_{\tilde{1}}$ gauge group breaks down, as the fractional magnetic charges of the constituent Dirac magnetic monopoles become observable at these scales. To make the fractional magnetic charges theoretically consistent, we need the full $U(1)_1 \times U(1)_2$ gauge group, as detailed in [19].

From Table 2, we see that while the constituent Dirac magnetic monopoles have magnetic charge $1/2$ in the broken $U(1)_{\tilde{2}}$ gauge group in our convention, the Higgs field has electric charge 2 in the same gauge group. This means that the magnetic flux tube has vorticity one.

The radius of the magnetic flux tube is of the order of $1/(gv)$. It is of the order the size of the MMM. So a slightly more realistic picture of the MMM considering the radius of the magnetic flux tube would look like Fig. 3. The schematic figure 2 may be compared with the structure formula in chemistry: it only captures how the constituent Dirac magnetic monopoles are connected by the magnetic flux tubes: it does not capture the actual configuration precisely, e.g. the angles between the magnetic flux tubes, nor the actual radius of the magnetic flux tubes.[3]

---

[3]The structure diagram in chemistry may better capture the actual configuration, after solving the Schrödinger equations for electrons. In principle, we can do better in these figures of MMMs, by solving the classical equations of motion.

| particle | charge in $U(1)_1 \times U(1)_2$ | charge in $U(1)_{\tilde{1}} \times U(1)_{\tilde{2}}$ |
|---|---|---|
| $\psi_1$ | (e) $(1,0)$ | (e) $(1,-1)$ |
| $\psi_2$ | (e) $(0,1)$ | (e) $(1,1)$ |
| Higgs $H$ | (e) $(-1,1)$ | (e) $(0,2)$ |
| $U(1)_1$ monopole | (m) $(1,0)$ | (m) $\left(\frac{1}{2},-\frac{1}{2}\right)$ |
| $U(1)_2$ monopole | (m) $(0,1)$ | (m) $\left(\frac{1}{2},\frac{1}{2}\right)$ |

Table 2: The case $N = 2$, $X = 1$, $Y = 1$: (e) and (m) indicate the electric- and magnetic charges, respectively. The charges are normalized so that the smallest charge units in the respective $U(1)$ gauge groups are one.

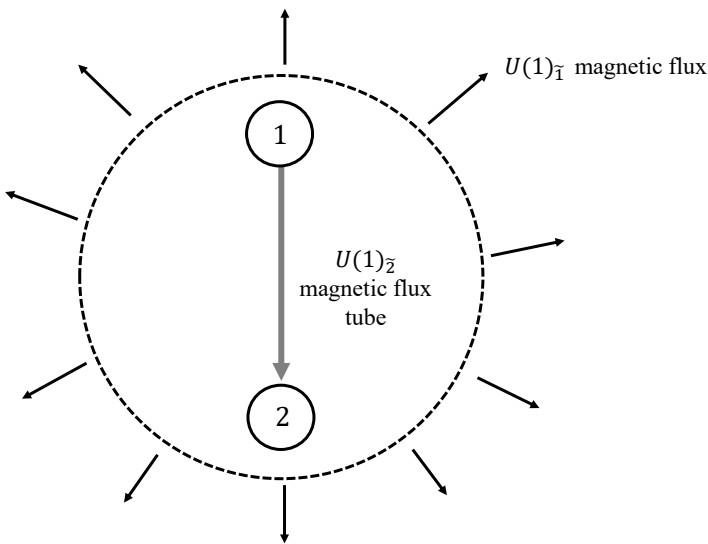

Figure 2: A schematic figure of the MMM for the case $X = 2$, $Y = 3$. The circles with 1 (2) represent the constituent Dirac magnetic monopoles in $U(1)_1$ ($U(1)_2$) gauge group. The broad arrow in grey indicates the magnetic flux tube of the broken $U(1)_{\tilde{2}}$ gauge group. Note that the actual configuration is in 3D. Also note that determining the precise locations of the constituent Dirac magnetic monopoles are beyond the scope of the current study; we are content with the order of magnitude estimate of the MMM. The outbound thin arrows indicate the magnetic flux of the unbroken $U(1)_{\tilde{1}}$ gauge group. From distances farther than the size of the MMM, it appears as a unit charge Dirac magnetic monopole.

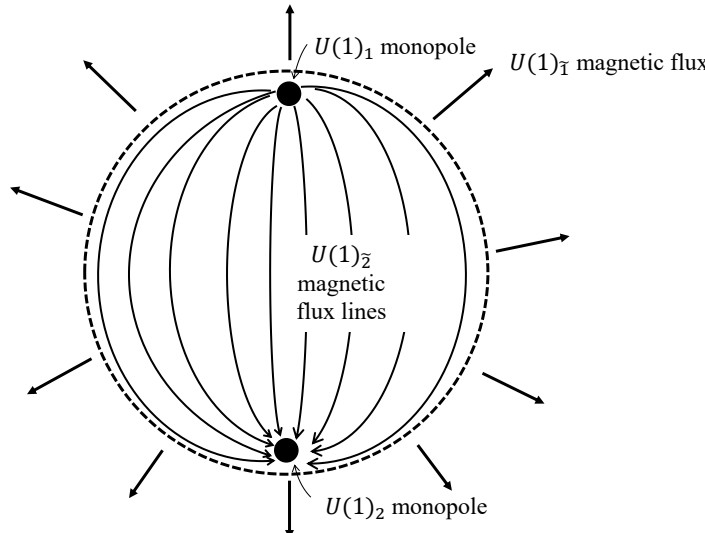

Figure 3: A figure slightly closer to the actual configuration than Fig. 3, in that the radius of the magnetic flux tube is considered.

### 2.1.2   $N = 2$, $X = 2$, $Y = 3$

We next study the case with $X = 2$, $Y = 3$ for illustrative purpose. The charges of the particles and the Dirac magnetic monopoles are listed in Table 3. Two Dirac magnetic monopoles of $U(1)_1$ and three Dirac magnetic monopoles of $U(1)_2$ constitute the MMM. The way magnetic flux tubes connect constituent Dirac magnetic monopoles is schematically depicted in Fig. 4.

| particle | charge in $U(1)_1 \times U(1)_2$ | charge in $U(1)_{\tilde{1}} \times U(1)_{\tilde{2}}$ |
|---|---|---|
| $\psi_1$ | (e) $(1, 0)$ | (e) $(2, -3)$ |
| $\psi_2$ | (e) $(0, 1)$ | (e) $(3, 2)$ |
| Higgs $H$ | (e) $(-3, 2)$ | (e) $(0, 13)$ |
| $U(1)_1$ monopole | (m) $(1, 0)$ | (m) $\left(\frac{2}{13}, -\frac{3}{13}\right)$ |
| $U(1)_2$ monopole | (m) $(0, 1)$ | (m) $\left(\frac{3}{13}, \frac{2}{13}\right)$ |

Table 3: The case $N = 2$, $X = 2$, $Y = 3$: (e) and (m) indicate the electric- and magnetic charge, respectively. The charges are normalized so that the smallest charge units in the respective $U(1)$ gauge groups are one.

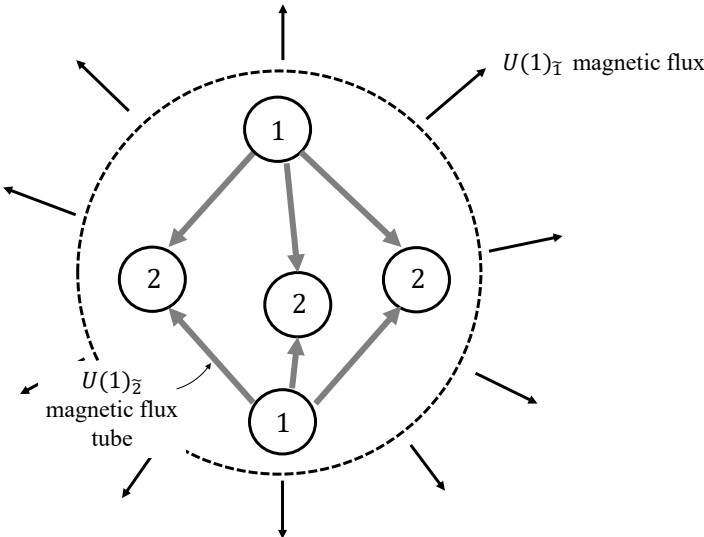

Figure 4: A schematic figure of the MMM for the case $N = 2$, $X = 2$, $Y = 3$. The circles with 1 (2) represent the constituent Dirac magnetic monopoles in $U(1)_1$ ($U(1)_2$) gauge group. The broad grey arrows indicate the magnetic flux tubes of the broken $U(1)_{\tilde{2}}$ gauge group. Note that the actual configuration is in 3D: Determining the precise locations of the constituent Dirac magnetic monopoles are beyond the scope of the current study; we are content with the order of magnitude estimate of the ground state of the MMM. The thin outbound arrows indicate the flux of the unbroken $U(1)_{\tilde{1}}$ gauge group.

## 2.2 $N = 3$

We consider a $U(1)_1 \times U(1)_2 \times U(1)_3$ product gauge group. We associate the ortho-normal basis vectors $\vec{e}_i$ to $U(1)_i$ gauge groups $(i = 1, 2, 3)$, respectively:

$$\vec{e}_i \cdot \vec{e}_j = \delta_{ij} \,. \tag{2.14}$$

We parametrize the unbroken $U(1)$ direction by three integers $(X, Y, Z)$ as follows:

$$\vec{e}_{\tilde{1}} = \sin\theta\cos\phi\,\vec{e}_1 + \sin\theta\sin\phi\,\vec{e}_2 + \cos\theta\,\vec{e}_3 \,, \tag{2.15}$$

where

$$\cos\theta = \frac{Z}{\sqrt{X^2 + Y^2 + Z^2}} \,, \quad \cos\phi = \frac{X}{\sqrt{X^2 + Y^2}} \,. \tag{2.16}$$

This is simple but not the most general possibility, but it covers a large class of possible configurations. More general MMMs can be obtained by parametrizing the charges of two Higgs fields, but they are slightly messier. Here, we are content with covering a large class of MMMs, leaving the more general case to future study.

The ortho-normal basis in broken directions is not unique. A convenient basis, though not the most general one, can be given as

$$\vec{e}_{\tilde{2}} = -\sin\phi\,\vec{e}_1 + \cos\phi\,\vec{e}_2 \,, \tag{2.17}$$

$$\vec{e}_{\tilde{3}} = \cos\theta\cos\phi\,\vec{e}_1 + \cos\theta\sin\phi\,\vec{e}_2 - \sin\theta\,\vec{e}_3 \,. \tag{2.18}$$

The inverse relation is given as

$$\vec{e}_1 = \sin\theta\cos\phi\,\vec{e}_{\tilde{1}} - \sin\phi\,\vec{e}_{\tilde{2}} + \cos\theta\cos\phi\,\vec{e}_{\tilde{3}} \,, \tag{2.19}$$

$$\vec{e}_2 = \sin\theta\sin\phi\,\vec{e}_{\tilde{1}} + \cos\phi\,\vec{e}_{\tilde{2}} + \cos\theta\sin\phi\,\vec{e}_{\tilde{3}} \,, \tag{2.20}$$

$$\vec{e}_3 = \cos\theta\,\vec{e}_{\tilde{1}} - \sin\theta\,\vec{e}_{\tilde{3}} \,. \tag{2.21}$$

In our convention that the smallest charge unit is one, the gauge couplings are normalized as

$$g_{\tilde{1}} = \frac{g}{\sqrt{X^2 + Y^2 + Z^2}} \,, \tag{2.22}$$

$$g_{\tilde{2}} = \frac{g}{\sqrt{X^2 + Y^2}} \,, \tag{2.23}$$

$$g_{\tilde{3}} = \frac{g}{\sqrt{(X^2 + Y^2 + Z^2)(X^2 + Y^2)}} \,. \tag{2.24}$$

Here, we assumed that $ZX$, $ZY$ and $X^2 + Y^2$ do not have a common divisor. This is not the case, for example, $X = 3$, $Y = 4$, $Z = 5$. The case $X = 3$, $Y = 4$, $Z = 5$ is separately studied below.

From Table 4, we observe that the collection of $X$ Dirac magnetic monopoles of $U(1)_1$, $Y$ Dirac magnetic monopoles of $U(1)_2$, and $Z$ Dirac magnetic monopoles of $U(1)_3$, has a unit magnetic charge in the unbroken $U(1)_{\tilde{1}}$ gauge group, and no net charge in the broken $U(1)_{\tilde{2}}$ and $U(1)_{\tilde{3}}$ gauge groups. This collection of the constituent Dirac magnetic monopoles makes up the MMM.

| particle | $U(1)_1 \times U(1)_2 \times U(1)_3$ | $U(1)_{\tilde{1}} \times U(1)_{\tilde{2}} \times U(1)_{\tilde{3}}$ |
|---|---|---|
| $\psi_1$ | (e) $(1,0,0)$ | (e) $(X,-Y,ZX)$ |
| $\psi_2$ | (e) $(0,1,0)$ | (e) $(Y,X,ZY)$ |
| $\psi_3$ | (e) $(0,0,1)$ | (e) $(Z,0,-(X^2+Y^2))$ |
| Higgs $H_{\tilde{2}}$ | (e) $(-Y,X,0)$ | (e) $(0,X^2+Y^2,0)$ |
| Higgs $H_{\tilde{3}}$ | (e) $(ZX,ZY,-(X^2+Y^2))$ | (e) $(0,0,(X^2+Y^2+Z^2)(X^2+Y^2))$ |
| $U(1)_{(1)}$ monopole | (m) $(1,0,0)$ | (m) $\left(\frac{X}{X^2+Y^2+Z^2},-\frac{Y}{X^2+Y^2},\frac{ZX}{(X^2+Y^2+Z^2)(X^2+Y^2)}\right)$ |
| $U(1)_{(2)}$ monopole | (m) $(0,1,0)$ | (m) $\left(\frac{Y}{X^2+Y^2+Z^2},\frac{X}{X^2+Y^2},\frac{ZY}{(X^2+Y^2+Z^2)(X^2+Y^2)}\right)$ |
| $U(1)_{(3)}$ monopole | (m) $(0,0,1)$ | (m) $\left(\frac{Z}{X^2+Y^2+Z^2},0,-\frac{1}{X^2+Y^2+Z^2}\right)$ |

Table 4: The case $N=3$: (e) and (m) indicate the electric- and magnetic charge, respectively. The charges are normalized so that the smallest charge units in the respective $U(1)$ gauge groups are one. Here, $X$, $Y$, $Z$ are mutually prime integers. We further assume that $ZX$, $ZY$ and $X^2+Y^2$ do not have a common divisor. This is not the case, for example, $X=3$, $Y=4$, $Z=5$: We separately list this case in Table 6.

### 2.2.1  $N=3$, $X=1$, $Y=1$, $Z=1$

We first study the simplest case $X=1$, $Y=1$, $Z=1$. Putting $X=1$, $Y=1$, $Z=1$ in (2.22)-(2.24), we obtain

$$g_{\tilde{1}} = \frac{g}{\sqrt{3}}, \quad g_{\tilde{2}} = \frac{g}{\sqrt{2}}, \quad g_{\tilde{3}} = \frac{g}{\sqrt{6}}. \tag{2.25}$$

The units of the magnetic charges in $U(1)_{\tilde{i}}$ gauge groups are given as $g_{m\tilde{i}} = 2\pi/g_{\tilde{i}}$ ($i=1,2,3$). From Table 5, we observe that one $U(1)_1$ Dirac magnetic monopole, one $U(1)_2$ Dirac magnetic monopole, and one $U(1)_3$ Dirac magnetic monopole constitute a unit charge Dirac magnetic monopole of the unbroken $U(1)_{\tilde{1}}$ gauge group. We also observe that the magnetic fluxes of the broken gauge groups $U(1)_{\tilde{2}}$ and $U(1)_{\tilde{3}}$ are confined within the MMM. From the charges of the Higgs fields in the broken gauge groups, we also observe that the magnetic flux tubes have an integer vorticity. A schematic figure of the MMM is given in Fig. 5.

| particle | $U(1)_1 \times U(1)_2 \times U(1)_3$ | $U(1)_{\tilde{1}} \times U(1)_{\tilde{2}} \times U(1)_{\tilde{3}}$ |
|---|---|---|
| $\psi_1$ | (e) $(1,0,0)$ | (e) $(1,-1,1)$ |
| $\psi_2$ | (e) $(0,1,0)$ | (e) $(1,1,1)$ |
| $\psi_3$ | (e) $(0,0,1)$ | (e) $(1,0,-2)$ |
| Higgs $H_{\tilde{2}}$ | (e) $(-1,1,0)$ | (e) $(0,2,0)$ |
| Higgs $H_{\tilde{3}}$ | (e) $(1,1,-2)$ | (e) $(0,0,6)$ |
| $U(1)_{(1)}$ monopole | (m) $(1,0,0)$ | (m) $\left(\frac{1}{3}, -\frac{1}{2}, \frac{1}{6}\right)$ |
| $U(1)_{(2)}$ monopole | (m) $(0,1,0)$ | (m) $\left(\frac{1}{3}, \frac{1}{2}, \frac{1}{6}\right)$ |
| $U(1)_{(3)}$ monopole | (m) $(0,0,1)$ | (m) $\left(\frac{1}{3}, 0, -\frac{1}{3}\right)$ |

Table 5: The case $N = 3$, $X = 1$, $Y = 1$, $Z = 1$: (e) and (m) indicate the electric- and magnetic charge, respectively. The charges are normalized so that the smallest charge units in the respective $U(1)$ gauge groups are one.

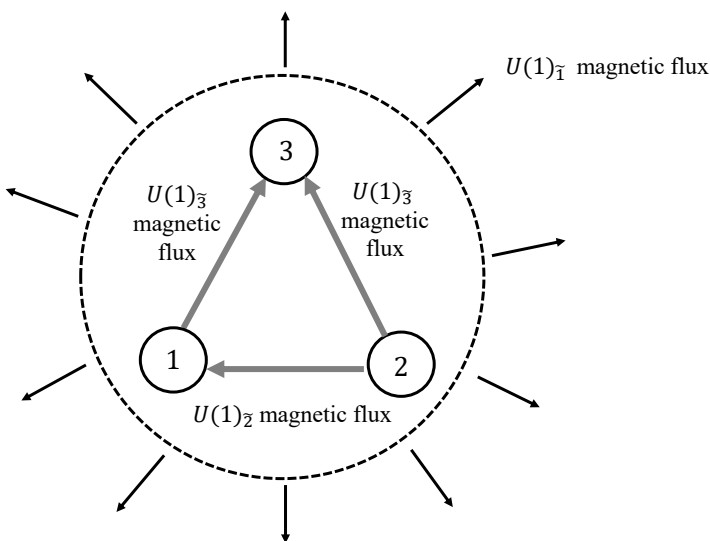

Figure 5: A schematic figure of the MMM for the case $N = 3$, $X = 1$, $Y = 1$. $Z = 1$. The circles with 1 (2, 3) represent the constituent Dirac magnetic monopoles in $U(1)_1$ ($U(1)_2$, $U(1)_3$) gauge group. The broad grey arrows indicate the magnetic flux tubes of the broken gauge groups.

### 2.2.2 $N = 3$, $X = 3$, $Y = 4$, $Z = 5$

Putting $X = 3$, $Y = 4$, $Z = 5$ in (2.22) and (2.23) while rescaling (2.24) by the common divisor 5 of $ZX$, $ZY$, $X^2 + Y^2$, we obtain

$$g_{\tilde{1}} = \frac{g}{\sqrt{50}} = \frac{g}{5\sqrt{2}}, \quad g_{\tilde{2}} = \frac{g}{\sqrt{25}} = \frac{g}{5}, \quad g_{\tilde{3}} = \frac{g}{\sqrt{50}} = \frac{g}{5\sqrt{2}}. \tag{2.26}$$

The units of the magnetic charges in $U(1)_{\tilde{i}}$ gauge groups are given as $g_{m\tilde{i}} = 2\pi/g_{\tilde{i}}$ ($i = 1, 2, 3$). From Table 6, we observe that three $U(1)_1$ Dirac magnetic monopoles, four $U(1)_2$ Dirac magnetic monopoles, and five $U(1)_3$ Dirac magnetic monopoles constitute a unit charge Dirac magnetic monopole of the unbroken $U(1)$ gauge group $U(1)_{\tilde{1}}$. We also observe that the magnetic fluxes of the broken gauge groups $U(1)_{\tilde{2}}$ and $U(1)_{\tilde{3}}$ are confined within the MMM. From the charges of the Higgs fields in the broken gauge groups, we also observe that the magnetic flux tubes have an integer vorticity.

As there are quite a few magnetic flux tubes in this MMM, the figure becomes somewhat messy. We leave the interested readers to try drawing the schematic figure in this case themselves.

| particle | $U(1)_1 \times U(1)_2 \times U(1)_3$ | $U(1)_{\tilde{1}} \times U(1)_{\tilde{2}} \times U(1)_{\tilde{3}}$ |
|---|---|---|
| $\psi_1$ | (e) $(1, 0, 0)$ | (e) $(3, -4, 3)$ |
| $\psi_2$ | (e) $(0, 1, 0)$ | (e) $(4, 3, 4)$ |
| $\psi_3$ | (e) $(0, 0, 1)$ | (e) $(5, 0, -5)$ |
| Higgs $H_{\tilde{2}}$ | (e) $(-4, 3, 0)$ | (e) $(0, 25, 0)$ |
| Higgs $H_{\tilde{3}}$ | (e) $(3, 4, -5)$ | (e) $(0, 0, 250)$ |
| $U(1)_1$ monopole | (m) $(1, 0, 0)$ | (m) $\left(\frac{3}{50}, -\frac{4}{25}, \frac{3}{50}\right)$ |
| $U(1)_2$ monopole | (m) $(0, 1, 0)$ | (m) $\left(\frac{4}{50}, \frac{3}{25}, \frac{4}{50}\right)$ |
| $U(1)_3$ monopole | (m) $(0, 0, 1)$ | (m) $\left(\frac{1}{10}, 0, -\frac{1}{10}\right)$ |

Table 6: The case $N = 3$, $X = 3$, $Y = 4$, $Z = 5$: (e) and (m) indicate the electric- and magnetic charge, respectively. The charges are normalized so that the smallest charge units in the respective $U(1)$ gauge groups are one.

## 2.3 Combining MMMs

A MMM appears as a Dirac magnetic monopole at distance scales much larger than its size. Then, we can use them as the constituent Dirac magnetic monopoles to construct larger MMMs. Note that we need to consider a gauge theory with a sufficient number of $U(1)$ gauge groups to combine two MMMs: If a MMM is realized in a $U(1)^{N_1}$ product gauge group which breaks down to a $U(1)$ subgroup, and another MMM is in a $U(1)^{N_2}$ product gauge group which breaks down to a $U(1)$ subgroup, we can combine these two MMMs in a gauge theory with a $U(1)^N$ product gauge group, with $N \geq N_1 + N_2$. As we can see from (2.13) that applies as long as the number of the $U(1)$ gauge groups and the number of the constituent Dirac magnetic monopoles remains of order one, the size of the MMM is determined by the VEV of the relevant Higgs field and (for simplicity) the common gauge coupling $g$. We can first construct MMMs of the size $1/(gv)$, where $g$ is the gauge coupling of the original gauge groups and $v$ is the VEV of the relevant Higgs field.

Let $\tilde{g}$ be the common gauge coupling of the unbroken $U(1)$ gauge groups at this scale. We consider Higgsing those gauge groups at this scale at larger distance scales. We take a product of the gauge groups to which the respective MMMs, which we use as constituent Dirac magnetic monopoles to construct larger MMMs, are magnetically charged. Then, from these constituent MMMs, we can construct a larger MMM of the size $1/(\tilde{g}w)$, where $w$ is the VEV of the relevant Higgs field such that $gv \gg \tilde{g}w$.

Now, consider bringing the scale $\tilde{g}w$ close to $gv$. In this process, the magnetic charges of the original constituent Dirac magnetic monopoles and the way how the magnetic fluxes of the broken gauge groups flow between them does not change. Therefore, we could actually combine MMMs at the same distance scale. Below, we explain the above process in simple examples.

### 2.3.1 Adding One Magnetic Monopole to the Diatomic MMMs

Actually, one of the Dirac magnetic monopoles to be added need not be MMM. Hence, we start with adding a Dirac magnetic monopole to the simplest MMM, a diatomic MMM of Sec. 2.1.1. We begin with the gauge group $U(1)_{11} \times U(1)_{12} \times U(1)_2$. For simplicity, we set the gauge couplings of the first two $U(1)$ gauge groups to be the same:

$$g_{11} = g_{12} = g \,. \tag{2.27}$$

See Table 7 for the charges of the fields and the magnetic monopoles. We also list the total magnetic charges of the intermediate and the final MMMs for convenience.

We first consider the gauge symmetry breaking of the $U(1)_{11} \times U_{12}$ gauge groups to $U(1)_{1\tilde{1}}$ by the Higgs field $H_{1\tilde{2}}$ acquiring the VEV. This part is exactly the same as the construction of the diatomic MMM we studied in Sec. 2.1.1. The gauge coupling of the unbroken $U(1)_{1\tilde{1}\tilde{1}}$ gauge group is given as

$$\tilde{g} = \frac{g}{\sqrt{2}} \,. \tag{2.28}$$

Next, for simplicity, we set the gauge coupling of the $U(1)_2$ gauge group to be $\tilde{g} = g/\sqrt{2}$, and consider the symmetry breaking $U(1)_{1\tilde{1}} \times U(1)_2 \to U(1)_{\tilde{1}}$.[4] The charge of the Higgs field $H_{\tilde{2}}$ is determined to realize this gauge symmetry breaking, while respecting the charge quantization conditions in the original gauge group $U(1)_{11} \times U(1)_{12} \times U(1)_2$, keeping the original smallest units of the electric charges.

---

[4]The spontaneous gauge symmetry breakings need not take place in a particular time order; this is merely the order in our sequence of the changes of the basis vectors in the space of the $U(1)$ gauge groups. In fact, while the spontaneous gauge symmetry breaking motivates us to choose a basis vector in the direction of the unbroken gauge group and the rest of the ortho-normal basis vectors, the change of the basis vectors itself can be done regardless of whether the gauge symmetry is broken or not. However, we can consider a cosmological scenario in which such an assembly of MMMs does occur in a time sequence. Also see the Summary and Discussion section.

| particle | $U(1)_{11} \times U(1)_{12} \times U(1)_2$ | $U(1)_{1\tilde{1}} \times U(1)_{1\tilde{2}} \times U(1)_2$ | $U(1)_{\tilde{1}} \times U(1)_{1\tilde{2}} \times U(1)_{\tilde{2}}$ |
|---|---|---|---|
| $\psi_{11}$ | (e) $(1,0,0)$ | (e) $(1,-1,0)$ | (e) $(1,-1,0)$ |
| $\psi_{12}$ | (e) $(0,1,0)$ | (e) $(1,1,0)$ | (e) $(1,1,0)$ |
| $\psi_2$ | (e) $(0,0,1)$ | (e) $(0,0,1)$ | (e) $(1,0,1)$ |
| Higgs $H_{1\tilde{2}}$ | (e) $(-1,1,0)$ | (e) $(0,2,0)$ | (e) $(0,2,0)$ |
| Higgs $H_{\tilde{2}}$ | (e) $(-1,-1,2)$ | (e) $(-2,0,2)$ | (e) $(0,0,4)$ |
| $U(1)_{11}$ monopole | (m) $(1,0,0)$ | (m) $\left(\frac{1}{2},-\frac{1}{2},0\right)$ | (m) $\left(\frac{1}{4},-\frac{1}{2},-\frac{1}{4}\right)$ |
| $U(1)_{12}$ monopole | (m) $(0,1,0)$ | (m) $\left(\frac{1}{2},\frac{1}{2},0\right)$ | (m) $\left(\frac{1}{4},\frac{1}{2},-\frac{1}{4}\right)$ |
| $U(1)_2$ monopole | (m) $(0,0,1)$ | (m) $(0,0,1)$ | (m) $\left(\frac{1}{2},0,\frac{1}{2}\right)$ |
| $U(1)_{1\tilde{1}}$ MMM | (m) $(1,1,0)$ | (m) $(1,0,0)$ | (m) $\left(\frac{1}{2},0,-\frac{1}{2}\right)$ |
| $U(1)_{\tilde{1}}$ MMM | (m) $(1,1,1)$ | (m) $(1,0,1)$ | (m) $(1,0,0)$ |

Table 7: (e) and (m) indicate the electric- and magnetic charge, respectively. We also list the total magnetic charges of the intermediate diatomic MMM and the final combined MMM. The charges are normalized so that the smallest charge units in the respective $U(1)$ gauge groups are one.

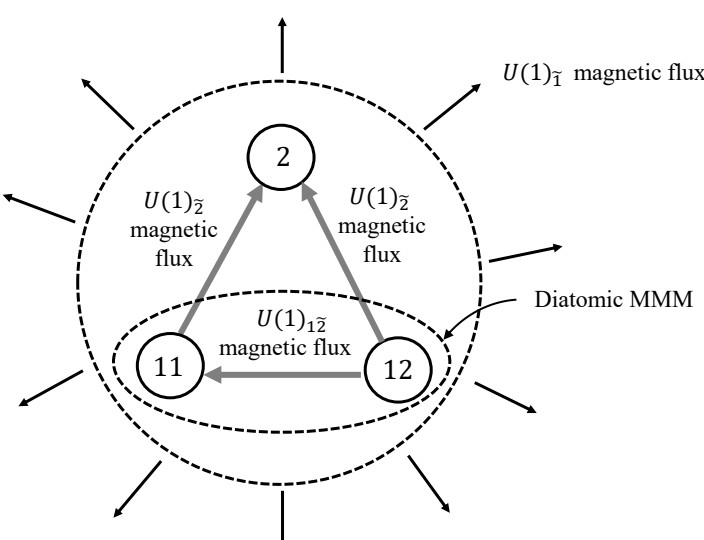

Figure 6: A schematic figure of the MMM constructed from adding a Dirac magnetic monopole to a diatomic MMM.

### 2.3.2 Combining Two Diatomic MMMs

Below, we describe how to combine two diatomic MMMs discussed in Sec. 2.1.1. For simplicity, we start with the gauge group $U(1)_{11} \times U(1)_{12} \times U(1)_{21} \times U(1)_{22}$ with a common

gauge coupling $g$. The explanations below should be read with Table 8. We also list the total magnetic charges of the intermediate diatomic MMMs for convenience. The magnetic charges are quantized in the unit

$$g_m := \frac{2\pi}{g} \, . \tag{2.29}$$

After (see the footnote 4) the Higgs fields $H_{1\tilde{2}}$ and $H_{2\tilde{2}}$ acquire vacuum expectation values, the gauge couplings in $U(1)_{1\tilde{1}}$, $U(1)_{1\tilde{2}}$, $U(1)_{2\tilde{1}}$, and $U(1)_{2\tilde{2}}$ become

$$\tilde{g} = \frac{g}{2} \, . \tag{2.30}$$

Consequently, the magnetic charges are quantized in the unit

$$\tilde{g}_m = \frac{2\pi}{\tilde{g}} = \frac{4\pi}{g} \, . \tag{2.31}$$

After the Higgs field $H_{\tilde{2}}$ acquires a vacuum expectation value, the gauge couplings become

$$\tilde{\tilde{g}} = \frac{\tilde{g}}{2} = \frac{g}{4} \, . \tag{2.32}$$

The magnetic charges are quantized in the unit

$$\tilde{\tilde{g}}_m = \frac{2\pi}{\tilde{\tilde{g}}} = \frac{8\pi}{g} \, . \tag{2.33}$$

The schematic figure of the MMM is given in Fig. 7.

| particle | Original | SSB1 | SSB2 |
|---|---|---|---|
| $\psi_{11}$ | (e) $(1,0,0,0)$ | (e) $(1,-1,0,0)$ | (e) $(1,-1,-1,0)$ |
| $\psi_{12}$ | (e) $(0,1,0,0)$ | (e) $(1,1,0,0)$ | (e) $(1,1,-1,0)$ |
| $\psi_{21}$ | (e) $(0,0,1,0)$ | (e) $(0,0,1,-1)$ | (e) $(1,0,1,-1)$ |
| $\psi_{22}$ | (e) $(0,0,0,1)$ | (e) $(0,0,1,1)$ | (e) $(1,0,1,1)$ |
| Higgs $H_{1\bar{2}}$ | (e) $(-1,1,0,0)$ | (e) $(0,2,0,0)$ | (e) $(0,2,0,0)$ |
| Higgs $H_{2\bar{2}}$ | (e) $(0,0,-1,1)$ | (e) $(0,0,0,2)$ | (e) $(0,0,0,2)$ |
| Higgs $H_{\bar{2}\bar{1}}$ | (e) $(-1,-1,1,1)$ | (e) $(-2,0,2,0)$ | (e) $(0,0,4,0)$ |
| $U(1)_{11}$ monopole | (m) $(1,0,0,0)$ | (m) $\left(\frac{1}{2},-\frac{1}{2},0,0\right)$ | (m) $\left(\frac{1}{4},-\frac{1}{2},-\frac{1}{4},0\right)$ |
| $U(1)_{12}$ monopole | (m) $(0,1,0,0)$ | (m) $\left(\frac{1}{2},\frac{1}{2},0,0\right)$ | (m) $\left(\frac{1}{4},\frac{1}{2},-\frac{1}{4},0\right)$ |
| $U(1)_{21}$ monopole | (m) $(0,0,1,0)$ | (m) $\left(0,0,\frac{1}{2},-\frac{1}{2}\right)$ | (m) $\left(\frac{1}{4},0,\frac{1}{4},-\frac{1}{2}\right)$ |
| $U(1)_{22}$ monopole | (m) $(0,0,0,1)$ | (m) $\left(0,0,\frac{1}{2},\frac{1}{2}\right)$ | (m) $\left(\frac{1}{4},0,\frac{1}{4},\frac{1}{2}\right)$ |
| $U(1)_{1\bar{1}}$ MMM | (m) $(1,1,0,0)$ | (m) $(1,0,0,0)$ | (m) $\left(\frac{1}{2},0,-\frac{1}{2},0\right)$ |
| $U(1)_{2\bar{1}}$ MMM | (m) $(0,0,1,1)$ | (m) $(0,0,1,0)$ | (m) $\left(\frac{1}{2},0,\frac{1}{2},0\right)$ |
| $U(1)_{\bar{1}\bar{1}}$ MMM | (m) $(1,1,1,1)$ | (m) $(1,0,1,0)$ | (m) $(1,0,0,0)$ |

Table 8: Original: charges in $U(1)_{11}\times U(1)_{12}\times U(1)_{21}\times U(1)_{22}$; SSB1: charges in $U(1)_{1\bar{1}}\times U(1)_{1\bar{2}}\times U(1)_{2\bar{1}}\times U(1)_{2\bar{2}}$; SSB2: charges in $U(1)_{\bar{1}\bar{1}}\times U(1)_{1\bar{2}}\times U(1)_{\bar{2}\bar{1}}\times U(1)_{2\bar{2}}$. (e) and (m) indicate the electric- and magnetic charge, respectively. We also list the total magnetic charges of the intermediate diatomic MMMs and the combined final MMM. The charges are normalized so that the smallest charge units in the respective $U(1)$ gauge groups are one.

## 2.4 Gravitational Stability of the MMMs and Implications to the WGC

The swampland conjectures are proposed criteria to select out EFTs that are not consistent with quantum gravity, or likely to be equivalently, string theory [15]. There have been a number of swampland conjectures proposed in the past. Their plausibility and usefulness vary. One approach to increase the plausibility of a swampland conjecture is to construct many examples in controlled string theory compactifications that satisfy the swampland constraints. Supersymmetry is often required to keep the calculations under theoretical control. As a result, some swampland conjectures are plausible but may only be applicable to the energy scales right below the string scale or supersymmetry breaking scale. On the other hand, some swampland conjectures are useful for constraining EFTs far below the string scale or supersymmetry scale, but may not be plausibly derived from string theory. Even if we can prove a swampland conjecture at the energy scale close to the string scale, there can be many things between the string scale to low low-energy

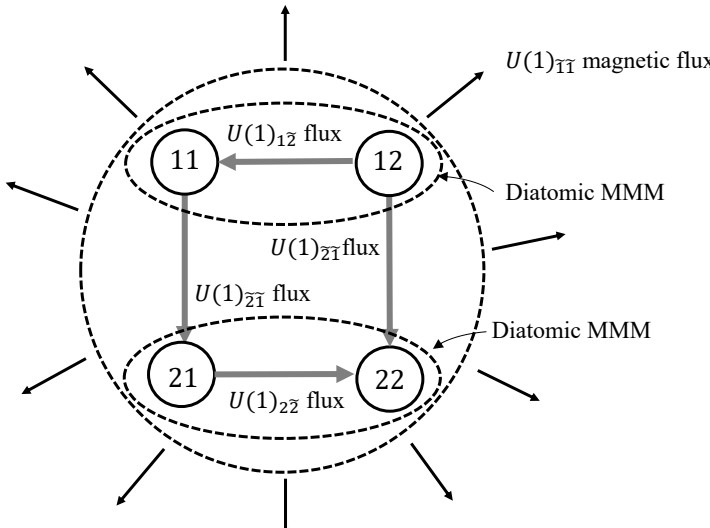

Figure 7: A schematic figure of the MMM constructed from combining two diatomic MMMs.

scales that might lead to a violation or an inapplicability of the swampland constraints, such as renormalization group flow, decoupling of heavy particles, spontaneous symmetry breakings, formation of composites, and so on. In order to make plausible swampland conjectures useful, we need to understand whether or how the constraints from the swampland conjectures descend to lower energy scales. Indeed, the MMM was first studied in this context [1], and successively investigated in [19, 25, 26, 27, 22].

The WGC [16] asserts that for EFTs with Abelian gauge symmetry, there exists a state for which the Coulomb repulsive force acts stronger than the Newtonian gravitational force:

$$gq \gtrsim \frac{m}{M_P}\,, \tag{2.34}$$

where $g$ is the gauge coupling, $q$ is the charge of the state, $m$ is the mass of the state, and $M_P$ is the reduced Planck mass. Requiring a similar condition for magnetically charged states, we obtain an upper bound for the UV cut-off scale $\Lambda_{\mathrm{UV}}$ above which the EFT breaks down:

$$\Lambda_{\mathrm{UV}} \lesssim gM_P\,. \tag{2.35}$$

The reason why the UV cut-off scale appears in (2.35) is that the mass of the Dirac magnetic monopoles are bounded from below as

$$m_{\mathrm{mono}} \gtrsim \frac{\Lambda_{\mathrm{UV}}}{g^2}\,. \tag{2.36}$$

The bound (2.35) can also be derived from the condition that the unit charge Dirac magnetic monopole should not be a black hole [16]: The size of the Dirac magnetic

monopole in an EFT is bounded from below by the UV cut-off $1/\Lambda_{\mathrm{UV}}$. On the other hand, the Schwarzschild radius for the mass $m_{\mathrm{mono}}$ is given as $\sim m_{\mathrm{mono}}/M_P^2$. Imposing that the size of the Dirac magnetic monopole is larger than the Schwarzschild radius, we obtain the bound (2.35). As we study the internal structure of the MMM, it is of interest to study the gravitational stability of the internal configuration of the MMM.

For concreteness, we consider the MMM with $N = 2$ studied in Sec. 2.1 with $X = 1$, which was essentially the same as the one originally studied by Saraswat [1], apart from minor conventional differences. See Table 9 for the charges of the particles and the constituent Dirac magnetic monopoles.

| particle | charge in $U(1)_1 \times U(1)_2$ | charge in $U(1)_{\tilde{1}} \times U(1)_{\tilde{2}}$ |
|---|---|---|
| $\psi_1$ | (e) $(1, 0)$ | (e) $(1, -Y)$ |
| $\psi_2$ | (e) $(0, 1)$ | (e) $(Y, 1)$ |
| Higgs $H$ | (e) $(-Y, 1)$ | (e) $(0, 1 + Y^2)$ |
| $U(1)_1$ monopole | (m) $(1, 0)$ | (m) $\left( \frac{1}{1+Y^2}, -\frac{Y}{1+Y^2} \right)$ |
| $U(1)_2$ monopole | (m) $(0, 1)$ | (m) $\left( \frac{Y}{1+Y^2}, \frac{1}{1+Y^2} \right)$ |

Table 9: The case $N = 2$, $X = 1$: (e) and (m) indicate the electric- and magnetic charge, respectively. The charges are normalized so that the smallest charge units in the respective $U(1)$ gauge groups are one.

The MMM in this case is composed of one Dirac magnetic monopole of the $U(1)_1$ gauge group and $Y$ Dirac magnetic monopole of the $U(1)_2$ gauge group. As in [1], we study $Y \gg 1$ case here, for a reason to become clear shortly. Note that in order to keep the theory in perturbative regime, we keep $gY$ fixed. Then, the gauge coupling of the unbroken gauge group $U(1)_{\tilde{1}}$ becomes (see (2.7))

$$\tilde{g} \simeq \frac{g}{Y} \,. \tag{2.37}$$

From this equation, when we have a large Higgs charge $Y \gg 1$, we have a very small gauge coupling $\tilde{g} \ll g$. This small gauge coupling raises a potential concern that the bound from the WGC (2.35) may be violated in the low-energy $U(1)_{\tilde{1}}$ gauge theory, even when the high-energy $U(1)_1 \times U(1)_2$ gauge theory respects the bound. It should be noted that the WGC bound the high-energy $U(1)_1 \times U(1)_2$ gauge theory and that on the low-energy $U(1)_{\tilde{1}}$ gauge theory are distinct [19]: the former talks about the $U(1)_1 \times U(1)_2$ gauge theory with the gauge coupling $g$, while the latter talks about $U(1)_{\tilde{1}}$ gaueg theory with the gauge coupling $\tilde{g}$. As mentioned earlier, the UV cut-off scale $\Lambda_{\mathrm{low}}$ of the latter is set by the size $L_{\mathrm{MMM}}$ (2.13) of the MMM [19].

The Dirac quantization condition leads to a larger magnetic charge unit for the unbroken gauge group $U(1)_{\tilde{2}}$:

$$\tilde{g}_m = \frac{2\pi}{\tilde{g}} \simeq \frac{2\pi Y}{g} = Y g_m \, . \tag{2.38}$$

Now, we study the gravitational stability of the internal structure of the MMM. In order for the MMM not to collapse, the attractive Newtonian gravitational potential should not surpass the repulsive magnetic Coulomb potential:

$$\frac{Y^2}{L} \left( \frac{\tilde{g}_m}{Y} \right)^2 \gtrsim \frac{Y^2}{L} \left( \frac{m_{\rm mono}}{M_P} \right)^2 \, , \tag{2.39}$$

where $L$ is the size of the configuration of the constituent Dirac magnetic monopoles spread. For the time being, we neglect the attractive forces due to the magnetic flux tubes, that is subdominant at the short scales. Putting (2.36) and (2.38) into (2.39), we obtain

$$\Lambda_{\rm UV} \lesssim g M_P \, . \tag{2.40}$$

This is nothing but the WGC bound (2.35) on the original $U(1)_1 \times U(1)_2$ gauge theory. Therefore, when the original $U(1)_1 \times U(1)_2$ gauge theory respects the WGC bound on the UV cut-off scale, the MMM does not gravitationally collapse.

As described in the earlier sections, the actual size of the MMM is determined by the balance between the repulsive magnetic Coulomb forces and the attractive forces from the magnetic flux tubes:

$$\frac{Y^2}{L_{\rm MMM}} \left( \frac{\tilde{g}_m}{Y} \right)^2 \simeq Y v^2 L_{\rm MMM} \, , \tag{2.41}$$

which leads to

$$L_{\rm MMM} \simeq \frac{\sqrt{Y}}{gv} \, . \tag{2.42}$$

As we fix $gY$ as we take $Y$ large to stay in a perturbative regime, it is convenient to rewrite (2.42) as

$$L_{\rm MMM} \simeq \frac{Y\sqrt{Y}}{(gY)v} \, . \tag{2.43}$$

In the meantime, the Schwarzschild radius of the configuration is given as[5]

$$L_{\rm BH} \simeq \frac{Y \Lambda_{\rm UV}}{g^2 M_P^2} \lesssim \frac{Y}{\Lambda_{\rm UV}} \, . \tag{2.44}$$

Here, we used the bound from the WGC (2.35) on the high-energy $U(1)_1 \times U(1)_2$ gauge theory. Comparing (2.43) and (2.44), we observe that 1. For the energy scales to start with, namely, $v$ and $\Lambda_{\rm UV}$, it is already natural to assume $v \ll \Lambda_{\rm UV}$: In order for the spontaneous gauge symmetry breaking to be described by the $U(1)_1 \times U(1)_2$ gauge theory,

---

[5]The mass energy of the constituent Dirac monopoles dominate the total energy of the MMM.

this inequality is required: $v$ is the VEV of the Higgs field, and unless the coupling parameters are fine-tuned, which is unnatural in the EFT framework, the inequality follows. 2. The $Y$ dependence further increase the hierarchy between $L_{\text{MMM}}$ and $L_{\text{BH}}$ at large $Y$, $L_{\text{MMM}} \gg L_{\text{BH}}$. This guarantees that the low-energy EFT with the $U(1)_{\bar{1}}$ gauge symmetry respects the WGC [19]:

$$\Lambda_{\text{low}} := \frac{1}{L_{\text{MMM}}} \lesssim \tilde{g} M_P \,. \tag{2.45}$$

These observations explain in some detail how the gravitational stability of the internal configuration of the MMM leads to the gravitational stability looked from outside the MMM. It provides an interesting case study for how the constraints from a swampland conjecture on a high-energy EFT descends to lower energy scales.

# 3  Summary and Discussions

In this article, we constructed a variety of configurations composed of Dirac magnetic monopoles connected by magnetic flux tubes in product $U(1)^N$ gauge theories, which we dubbed "Magnetic Monopole Molecules" (MMMs). The MMMs had the unit magnetic charge in the unbroken $U(1)$ gauge group, determined by the Dirac quantization condition. The configurations were stabilized by the balance between the repulsive magnetic Coulomb forces of the unbroken $U(1)$ gauge group and the attractive forces from the tension of the magnetic flux tubes of the broken $U(1)$ gauge groups. We presented how to combine MMMs to construct a larger MMM. We also studied the gravitational stability of the MMMs. It demonstrated how the constraints from the WGC on a high-energy EFT descend to lower energy scales in an explicit example. This will be useful for understanding better how a swampland conjecture can be both plausible and useful for constraining EFTs at energy scales far below the string scale or supersymmetry breaking scale.

Since $U(1)$ gauge (sub)groups are ubiquitous in grand unified theories and string theory compactifications, we expect that MMMs studied in this article have a good chance to play a role in our universe. As mentioned in Sec. 2.3, MMMs can have hierarchical structures; smaller MMMs can be produced at higher energy scales and larger MMMs can be produced at lower energy scales. Such a hierarchy may have interesting applications in cosmological phase transitions that might occur at different energy scales in different epochs of the universe.

### Acknowledgments

This work was supported in part by the project file no. DST/INT/JSPS/P-344/2022. We thank the participating members and the supporting staff, in particular, the PI of the JSPS side Keisuke Izumi for various arrangements. KF thanks Toshifumi Noumi for the

hospitality during his visit to the University of Tokyo, Komaba, supported by the above project, and for useful discussions. MP acknowledges Dr. T.M.A. Pai Ph.D. scholarship programme, Manipal Academy of Higher Education (MAHE). We thank Koushik Dutta and Anshuman Maharana for advice and encouragement on the Ph.D. project. Manipal Centre for Natural Sciences, *Centre of Excellence*, Manipal Academy of Higher Education (MAHE) is acknowledged for facilities and support.

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
