# Peer review of "Magnetic Monopole Molecules"

_SciPost Physics_

## Round 2 · Referee Report · Anonymous (Referee 1) · 2025-11-18

Strengths

1- Suggests the systematic analysis of bound states of monopoles in Higgsed theories. 2- Describes an approach for producing more complicated bound state solutions from simpler ones.

Weaknesses

1- No discussion of kinetic mixing. 2- Only considers equal gauge couplings. 3- Only fully analyzes the case of 2 monopoles. 4- Full analysis of stability is not carried out. 5- Discussion of implications for the (magnetic) WGC is essentially identical to that of 1608.06951.

Report

The general research question being asked is interesting and presents an opportunity to connect various avenues of research. There are nicely presented examples and clear discussion of the approach and theory. The paper does not present much analysis beyond what has been previously accomplished, particularly in the primary reference (1608.06951). In particular the paper does not address the significant obstacle of generalizing to theories with kinetic mixing, or explain why such a generalization should be straightforward. Due to the limited number of completely worked examples, minimal general theory, and redundant connection to the magnetic WGC, I cannot currently recommend this paper for publication. I request the authors extend their work to include kinetic mixing or explain how this is accomplished, discuss in more detail what the stable configuration of the monopoles is, and expand their discussion of connections to the magnetic WGC/swampland program to encompass novel results.

Requested changes

1- Explain how to construct composite monopoles in theories with kinetic mixing. 2- Explicitly work out more examples: when there are more than 2 component monopoles the flux tube structure and effective size should be discussed. 3- Explicitly demonstrate the stability of these composite monopoles: the force balance argument is well known, but how this organizes into a (meta)stable state is warranted. 4- Expand analysis of these composite monopoles role in the magnetic WGC/swampland to present novel results.

Recommendation

Ask for major revision

---

## Editorial Decision

awaiting_resubmission